# Optimal Control of Chilled Water System Based on Improved Sparrow Search Algorithm

**Qixin Zhu** [1,2,*], **Mengxiang Zhuang** [3], **Hongli Liu** [1] and **Yonghong Zhu** [4]

1   School of Mechanical Engineering, Suzhou University of Science and Technology, Suzhou 215009, China; liuhl_sz@163.com
2   Jiangsu Province Key Laboratory of Intelligent Building Energy Efficiency, Suzhou University of Science and Technology, Suzhou 215009, China
3   School of Environmental Science and Engineering, Suzhou University of Science and Technology, Suzhou 215009, China; mzhuang@mail.usts.edu.cn
4   School of Mechanical and Electronic Engineering, Jingdezhen Ceramic Institute, Jingdezhen 333001, China; zyh_patrick@163.com
*   Correspondence: qxzhu@mail.usts.edu.cn

**Abstract:** Chilled water systems have large time delays and large inertia, and the traditional PID controller has a poor control effect. In this paper, an improved sparrow search algorithm is proposed to optimize the control of chilled water systems. Firstly, the random walk strategy was used to randomly perturb the sparrows to improve the searching ability of the sparrows. Then, a Gauss mutation was added in the iteration process of sparrows to enhance the local search ability. Finally, the values of the PID parameters as obtained by the above methods were substituted into the controller for simulation. The simulation results show that the method proposed in this paper improves the search accuracy of the sparrow search algorithm and effectively solves the problems of large time delays and large inertia in the chilled water system. The method in this paper took the least amount of time for the system to reach the steady state at only 12.75 s. The control effect of the proposed method was also better than that of the improved ant colony optimization algorithm. The rise time was 2.713 s, and the adjustment time was 4.95 s.

**Keywords:** sparrow search algorithm; random walk strategy; Gauss mutation; PID parameter optimization; chilled water system

## 1. Introduction

A chilled water system is the main part of an HVAC system. It consumes a significant amount of energy and has an important influence on the refrigeration effect [1]. The energy consumption of a chilled water system comprises about 30% of the energy consumption of air conditioning [2,3]. In a chilled water system, energy efficiency can be achieved by changing the equipment, control strategy, pipeline layout [4] or temperature control [5] or using other methods. A chilled water system is time-varying and has a time delay [6]. A chilled water system with a better control effect would be conducive to creating energy-saving air conditioning systems [7]. Z.J. Ma et al. proposed an online control scheme. Compared with the conventional control method, the energy consumption of the chilled water pump was reduced by 11.99–24.86% [8]. Using the traditional PID controller to control the system will lead to the system to overshoot, resulting in slow responses and other problems. It does not work very well.

Swarm optimization algorithm has been applied in various fields, including ant colony optimization (ACO) [9], particle swarm optimization (PSO) [10], grey wolf optimization (GWO) [11] and so on. The above group optimization algorithms have the problem that they are easy to fall into the local optimum. Many people have improved the optimization algorithm. Y. Wang et al. used the improved ant colony algorithm to find the global optimal

solution [12]. F. Zhang et al. used Gauss mutation and self-adjusting pheromones to improve the ant colony algorithm to optimize the controller in a chilled water system and improved the searching ability and convergence ability of the algorithm [13]. A. Dixit et al. proposed a new differential evolution algorithm based on particle swarm optimization. The performance comparison showed that the algorithm could improve the convergence speed significantly and avoid premature closing [14]. J.C. Gu et al. added discrete heredity into the grey wolf optimization algorithm to further improve the search ability. The experimental data show that this algorithm was more effective than other algorithms [15]. J.K. Xue et al. proposed the Sparrow Search Algorithm (SSA) in 2020, which had good convergence speed and stability [16]. X.C. Li et al. used the SSA, which can quickly and accurately search the required parameters [17]. X.L. Sun et al. proposed an improved sparrow algorithm and applied it to load forecasting [18,19].

In order to achieve a better control effect, this paper used the sparrow search algorithm to find the optimal PID control parameters. However, like other algorithms, the sparrow search algorithm is also prone to fall into local optimum and population diversity decreases with the increase in iteration times. Based on the above description, in order to solve the disadvantages of the sparrow search algorithm, this paper proposes an improved sparrow search algorithm and used it to optimize the control of a chilled water system. Firstly, the random walk strategy was used to perturb the position of the sparrows to enhance the global search ability of the sparrow. Secondly, Gauss mutation was applied to change the sparrows' position, so that individual sparrows could carry out a full search of their surroundings, and the local searching ability of the sparrow group was improved. Finally, the optimization algorithm was applied to a chilled water system and compared with the ant colony algorithm and particle swarm optimization algorithm. The simulation results show that the proposed method improved the search precision of the sparrow search algorithm and reduced the overshoot of the system. The control effect was also better than that of other optimization algorithms. The main contributions of the authors are as follows: (1) The sparrow search algorithm was improved using the random walk strategy and Gaussian variation in this paper. (2) The improved sparrow search algorithm was applied to a frozen water system.

The paper is divided into five sections including the introduction. Section 2 presents the related methods. Section 3 presents the main results of this paper. A discussion is given in Section 4. The conclusion is presented in Section 5.

## 2. Related Methods

### 2.1. Sparrow Search Algorithm Optimizes PID Parameters

The sparrow search algorithm optimizes the values of the three parameters of the PID controller, which are proportional, integral and differential, and its purpose is to find the corresponding PID parameters when the objective function value is the lowest, which can control the control system more effectively. In the sparrow swarm algorithm, the finders are responsible for finding the optimal region of the objective function values, and the participants will move close to them to form an orderly sparrow swarm [16]. Participants who are in very bad positions will go to other areas to search. The participants will constantly monitor the finders through the position update and the feedback of the objective function values. If the participants' positions are better than those of the finders at this time, they will become the finders.

We sorted the objective function values from smallest to largest and updated the positions of each sparrow in real time. Finders account for 20% of the sparrows, and the rest are participants. The position update of the finders is shown in Equation (1).

$$x_{i,j}^{d+1} = \begin{cases} x_{i,j}^d \cdot e^{\left(\frac{-j}{\alpha \cdot dmax}\right)}, & \text{if } R_2 < ST \\ x_{i,j}^d + Q \cdot L, & \text{if } R_2 \geq ST \end{cases} \tag{1}$$



where $x_{i,j}^d$ represents the position of the $j$th sparrow in the $i$th dimension in the $d$th iteration, $i = 1, 2, 3, j = 1, 2, 3, \ldots, n, d = 1, 2, 3, \ldots, d_{max}$; $d_{max}$ represents the maximum number of iterations; $\alpha$ is a random number, and its value range is $(0, 1)$; $Q$ is a random number subject to normal distribution; $L$ represents a matrix of one row and three columns in which the elements are 1; $R_2$ represents the alert value, which is in the range of $(0, 1)$; $ST$ represents the safe value, and the value range is $(0.5, 1)$; When $R_2 < ST$, this indicates that the sparrows have not found the predator, the population is in a safe state and the finders can perform a more extensive search, thus finding a better feeding area. When $R_2 \geq ST$, this indicates that some sparrows have spotted a predator and will alert others, who will immediately change their search strategy and move to a safe area.

The number of participants accounts for 80% of the sparrows, and the position update is shown in Equation (2).

$$x_{i,j}^{d+1} = \begin{cases} Q \cdot e^{\left(\frac{x_{worst}^d - x_{i,j}^d}{j^2}\right)}, & \text{if } \quad j > \frac{n}{2} \\ x_p^d + \left|x_{i,j}^d - x_p^d\right| \cdot A^+ \cdot L, & \text{otherwise} \end{cases} \tag{2}$$

where $x_{worst}^d$ represents the worst position in the group; $x_p^t$ represents the best position occupied by the finders; A represents a matrix of one row and three columns in which the elements are randomly assigned to either $-1$ or 1, and $A^+ = A^T(AA^T)^{-1}$. When $j > n/2$, this indicates that the $j$th participant has a relatively low fitness value. It does not acquire food and is in a state of extreme hunger. It needs to fly somewhere else to find food. When $j \leq n/2$, this indicates that the $j$th participant will search for things around the best location found so far.

In the sparrows, the detection and early warning mechanism was also added, so that when the sparrow senses danger, and it will move to a safe place. In this paper, these sparrows account for 20% of the population, and the position update is shown in Equation (3).

$$x_{i,j}^{d+1} = \begin{cases} x_{best}^d + \beta \cdot \left|x_{i,j}^d - x_{best}^d\right|, & \text{if } \quad f_j > f_g \\ x_{i,j}^d + K \cdot \left(\frac{\left|x_{i,j}^d - x_{worst}^d\right|}{(f_i - f_w) + \varepsilon}\right), & \text{if } \quad f_j = f_g \end{cases} \tag{3}$$

where $x_{best}^d$ is the best position in the group; $\beta$ represents the step size control parameter and is a normal random number with a mean of 1 and a variance of 0. $K$ is a random number with the value range of $[-1, 1]$. $f_j$ is the fitness value of the $j$th sparrow; $f_g$ and $f_w$ represent the maximum and minimum fitness values in the population, respectively; and $\varepsilon$ is a very small constant. It keeps the denominator from being 0, and it is $10^{-8}$. When $f_j > f_g$, this indicates that sparrows are on the fringes of the population and are extremely vulnerable to predators. When $f_j \neq f_g$, this indicates that the sparrows in the middle of the group are aware of the danger and need to move closer to other sparrows to reduce the risk of predation.

Equations (1)–(3) are the updated formulae for the sparrow group position in the sparrow search algorithm. Parameters of the PID controller in the chilled water system are expressed in the form of a sparrow group, as shown in Equation (4).

$$X = \begin{bmatrix} x_{1,1}^d & x_{2,1}^d & x_{3,1}^d \\ x_{1,2}^d & x_{2,2}^d & x_{3,2}^d \\ \vdots & \vdots & \vdots \\ x_{1,n}^d & x_{2,n}^d & x_{3,n}^d \end{bmatrix} \tag{4}$$

where $d$ represents the number of current iterations; and $n$ represents the number of sparrows in the population.

The fitness function is shown in Equation (5).

$$f = \int_0^t e^2(t)\mathrm{d}t \tag{5}$$

where $e(t)$ is the difference between the input value and the output value.

In the group of sparrows, each column represents three parameters of PID. The fitness function value of each sparrow is shown in Equation (6).

$$F = \begin{bmatrix} f_1^d \\ f_2^d \\ \vdots \\ f_n^d \end{bmatrix} \tag{6}$$

*2.2. The Main Improvement Steps of Sparrow Search Algorithm*

2.2.1. Random Walk Strategy

The expression of the random walk strategy is shown in Equation (7).

$$X(t) = [0, cumsum(2r(rand(t,1)) - 1)] \tag{7}$$

where $X(t)$ is the set of steps of the random walk; *cumsum* is the formula for calculating the sum; and $t$ is the number of steps of the random walk, and it is $d_{max}$. $r(t)$ is a random number, and its definition is shown in Equation (8).

$$r(t) = \begin{cases} 1, & rand(t,1) > 0.5 \\ 0, & rand(t,1) \le 0.5 \end{cases} \tag{8}$$

where $rand(t,1)$ is a matrix of $t$ rows and 1 column, with the value range of [0, 1].

After updating the positions of the sparrows, the boundary values of the variables are updated, following the rule that the larger the number of iterations is, the smaller the scope of search is. The form is shown in Equations (9) and (10).

$$lb_i^{d+1} = \frac{-lb_i^d}{I} + x_{best}^d \tag{9}$$

$$ub_i^{d+1} = \frac{ub_i^d}{I} + x_{best}^d \tag{10}$$

where $lb_i^d$ represents the lower boundary value of the $i$th dimensional variable in the $d$th iteration; $ub_i^d$ represents the upper boundary value of the $i$th dimensional variable in the $d$th iteration; and $i$ represents the boundary reduction factor, as shown in Equation (11).

$$I = \begin{cases} 1 + 10^2 \cdot \frac{d}{d_{\max}}, & \text{if} \quad d > d_{max} \cdot 0.2 \\ 1 + 10^3 \cdot \frac{d}{d_{\max}}, & \text{if} \quad d > d_{max} \cdot 0.5 \\ 1 + 10^4 \cdot \frac{d}{d_{\max}}, & \text{if} \quad d > d_{max} \cdot 0.7 \\ 1 + 10^5 \cdot \frac{d}{d_{\max}}, & \text{if} \quad d > d_{max} \cdot 0.9 \\ 1 + 10^6 \cdot \frac{d}{d_{\max}}, & \text{if} \quad d > d_{max} \cdot 0.95 \end{cases} \tag{11}$$

After the random walk, the sparrows' positions are updated, as shown in Equation (12).

$$x_i^d = \frac{(x_i^d - a_i) \cdot (ub_i^d - lb_i^d)}{(b_i - a_i)} + lb_i^d \tag{12}$$

where $x_i^d$ represents the position of the $i$th dimension in the $d$th iteration of the sparrow; $a_i$ is the minimum value of the random walk of the $i$th dimensional variable; $b_i$ is the maximum value of the random walk of the $i$th dimensional variable.

### 2.2.2. Gauss Mutation

Gauss mutation uses the mutation factor of the genetic algorithm for reference. Gauss mutation was used to change the positions of sparrows in this paper. The mutation will produce a random number conforming to the normal distribution with a mean of $\mu$ and a standard deviation of $\sigma$. The fitness will be calculated according to the mutated value, and the original value will be chosen to be replaced. If the fitness value after the mutation is smaller than the value before the mutation, the original value will be replaced with the value after the mutation, and vice versa. The formula of the gauss mutation is shown in Equation (13).

$$x_i^d = x_i^d(1 + N(0,1)) \tag{13}$$

According to the characteristic of normal distribution, Gauss mutation has strong local search ability and can search the local area around sparrows adequately. The Gauss mutation improved the diversity of the sparrow search algorithm, which is conducive to finding the optimal position more quickly and accurately.

## 3. Results

### 3.1. Establishment of Optimization Model

The simulation of this paper was carried out in MATLAB 2019a. The mathematical model of the chilled water system is very complicated and belongs to a high-order system. Thus, a second-order model with time delay was used instead. The mathematical model of the chilled water system is shown in Equation (14).

$$G(s) = \frac{Ke^{-\tau s}}{(T_1 s + 1)(T_2 s + 1)} \tag{14}$$

where $T_1$ and $T_2$ are the inertial time constants; $K$ is the amplification factor; and $\tau$ is the lag time parameter of the chilled water system.

A block diagram of the optimized chilled water system based on the improved sparrow search algorithm is shown in Figure 1.

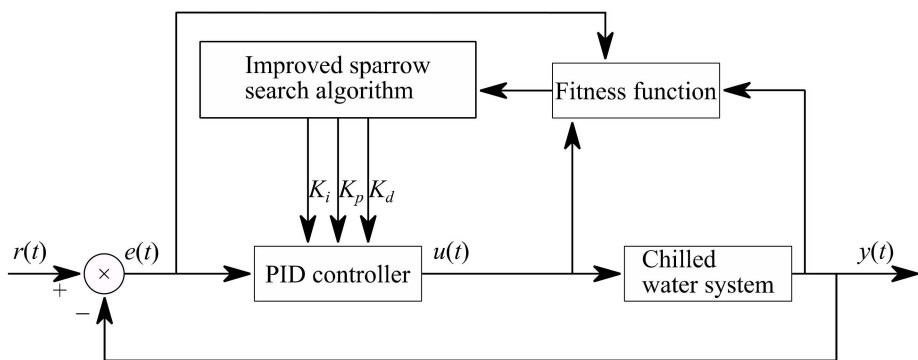

**Figure 1.** The block diagram of the chilled water system based on the algorithm.

### 3.2. Comparison of Simulation Effects

In order to verify the effectiveness of the proposed method, the proposed method was compared with the sparrow search algorithm, particle swarm optimization algorithm and ant colony optimization algorithm. The control method adopted was temperature difference control. The temperature difference between the supply and return water of the chilled water was set at 5 °C. The sampling time of the system was 0.5 s. The three parameters of the PID controller had an upper limit of 5 and a lower limit of 0. The group size of the swarm optimization algorithm in this paper was 50, and the maximum number of iterations was 100. Other parameters of the group algorithm were set as follows:

Particle swarm optimization algorithm: $w = 0.7$ (inertia factor) and $c1 = 2$, $c2 = 2$ (acceleration constant).

Ant colony optimization algorithm: *Rho* = 0.7 (Pheromone evaporation coefficient), *Q* = 1 (Pheromone intensity) and *Lam* = 0.2 (Crawling speed of ants).

The specific parameter values of the improved part and the sparrow search algorithm are detailed in Section 2.

The control object model adopted in this paper is shown in Equation (15). The optimal PID parameter values found by each algorithm are shown in Table 1. The simulation diagram of the system is shown in Figure 2.

**Table 1.** The optimization results of each algorithm.

| Methods | $K_p$ | $K_i$ | $K_d$ |
| --- | --- | --- | --- |
| This paper | 3.6982 | 0.034355 | 3.3944 |
| SSA | 1.4747 | 0.027479 | 0.44203 |
| PSO | 4.2188 | 0.0326 | 5 |
| ACO | 3.8493 | 0.0339 | 3.7885 |

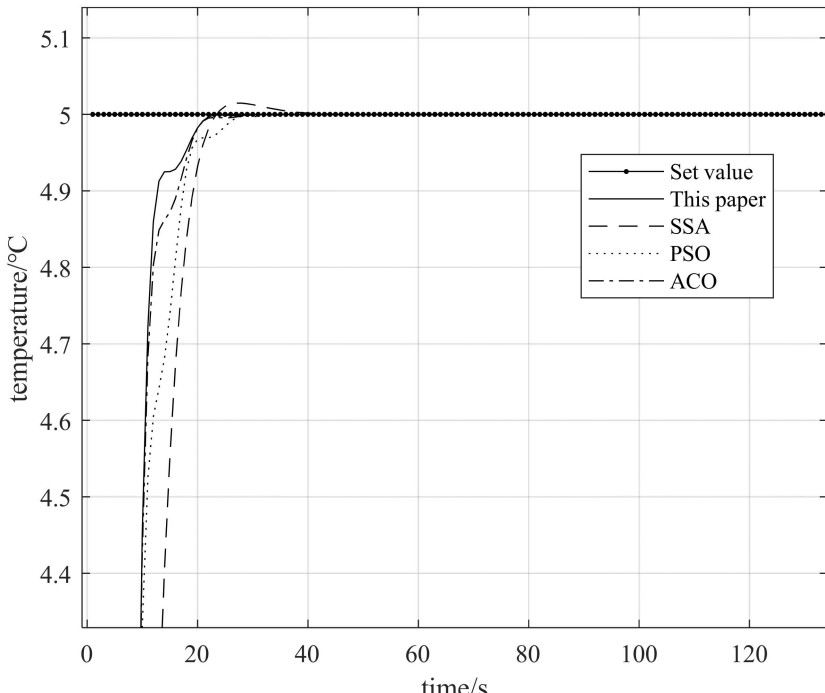

**Figure 2.** Simulation diagram of the system.

Since the application of the sparrow search algorithm in frozen water system has not been reported, in order to be more convincing, this paper also makes a comparison with literature. In Ref. [13], F. Zhang et al. used the improved ant colony algorithm to optimize the control of a chilled water system. In order to improve the searching ability and fast convergence, Gaussian variation and self-adjusting pheromones were introduced, and the sum of deviation squares was used as the objective function. The control method adopted was temperature difference control, which was controlled at 5 °C, and the outlet temperature of the chilled water was set at 7 °C. The purpose of the control was to keep the temperature of the return water at 12 °C. The model adopted by the control object is shown in Equation (15).

$$G(s) = \frac{12e^{-30s}}{(50s + 1)(s + 1)} \tag{15}$$

The sampling time of the system was 5 s, and the range of optimization parameters was $K_p \in [0, 0.6]$, $K_i \in [0, 0.5]$, $K_d \in [0, 1]$. The PID parameters optimized by the improved ant colony algorithm in [13] were 1.6561, 0.0325 and 0.8839. The PID parameters optimized

by the method in this paper were 0.26399, 0.0052043 and 0.83927. The simulation results are shown in Figure 3. The control performances of the two methods are shown in Table 2.

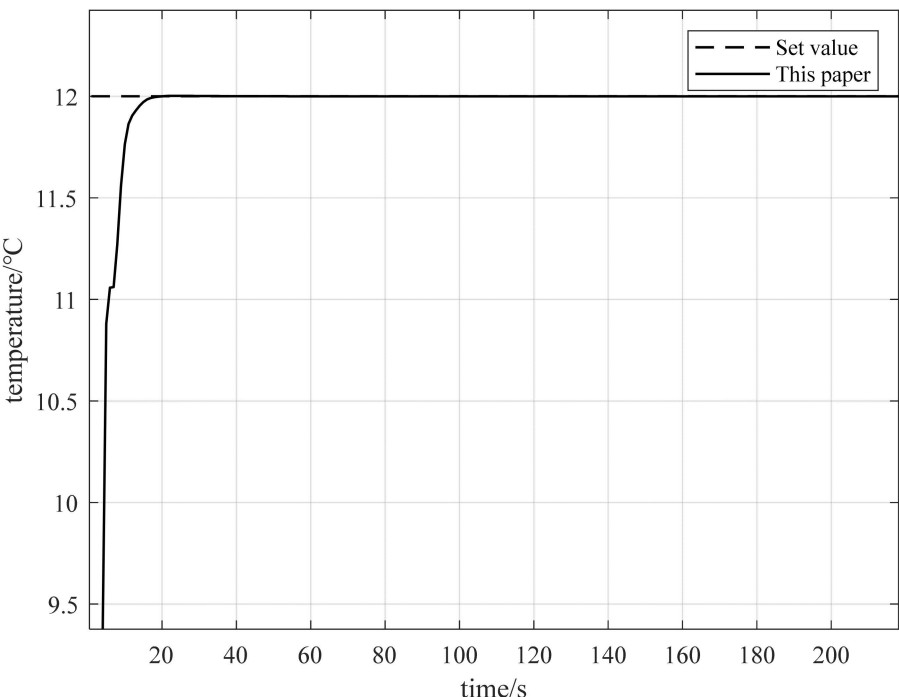

**Figure 3.** Simulation diagram of improved sparrow search algorithm.

**Table 2.** Comparison of control performance.

| Methods | Overshoot | Rise Time | Adjustment Time |
|---|---|---|---|
| Literature [13] | 0% | 4.4 s | 6.23 s |
| This paper | 0% | 2.713 s | 4.95 s |

## 4. Discussion

As can be seen from Figure 2, the simulation effect of the unimproved sparrow search algorithm was the worst, and even overshoot appeared. After the improvement in this paper, the overshoot was reduced. The method in this paper took the least amount of time for the system to reach the steady state at only 12.75 s. The time taken for the SSA, PSO and ACO to reach the steady state was 19.21 s, 17.45 s and 16.39 s, respectively. The proposed method has a better control effect than other methods.

From the perspective of control performance, the method proposed in this paper performed better than the control in [13] in both the rise time and adjustment time. The rise time was 2.713 s, and the adjustment time was 4.95 s. The control method proposed in this paper has a fast response speed, and its stability was improved accordingly. In view of the hysteresis of chilled water system, this method can improve the response speed of the controller.

## 5. Conclusions

In this paper, an improved sparrow search algorithm was proposed to optimize the control of chilled water systems. The random walk strategy was used to perturb the sparrows' positions, and Gauss mutation was added to improve the sparrows' search ability. Compared with the particle swarm optimization algorithm and ant colony optimization algorithm, the simulation results show that the improved method proposed in this paper not only improved the search ability of the sparrow search algorithm but also improved the control effect of the chilled water system. The method in this paper took the least

amount of time for the system to reach the steady state at only 12.75 s. The control effect of the proposed method is also better than that of the improved ant colony optimization algorithm. The rise time was 2.713 s, and the adjustment time was 4.95 s. In future research, we will apply other methods to the swarm optimization algorithm to further improve the search ability of the swarm optimization algorithm.

**Author Contributions:** Methodology, Q.Z. and M.Z.; investigation, Q.Z.; writing—original draft preparation, M.Z.; writing—review and editing, Q.Z., M.Z. and H.L.; software, M.Z. and H.L.; validation, Q.Z. and H.L.; data curation, M.Z. and Y.Z.; supervision, Q.Z. and H.L. All authors have read and agreed to the published version of the manuscript.

**Funding:** This research was partially supported by the National Nature Science Foundation of China (Grant No.51875380 and 62063010).

**Data Availability Statement:** The data presented in this study are available on request from the corresponding author.

**Acknowledgments:** We acknowledge any support given which is not covered by the author contribution or funding sections.

**Conflicts of Interest:** We declare no conflict of interest.

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
