# Peer review of "Optimal Control of Chilled Water System Based on Improved Sparrow Search Algorithm"

_buildings, doi:10.3390/buildings12030269_

Round 1

Reviewer 1 Report

This paper (buildings-1573715) titled Optimal control of chilled water system based on improved 2 sparrow search algorithm presented details of a modified search algorithm meant to be used in the control of chilled water temperature.

According to the review major faults of the paper are:

  1. Literature should include more titles that deal with the development of the Sparrow search algorithm (especially similar to given research). The authors mentioned several, but it should be extended with more recent work.
  2. The introduction must include a more detailed and reasoned description to prove the novelty of this paper.
  3. Line 74. ..three parameters… they are given in the paper, but they should be explained here for easier reading and understanding.
  4. Chapter 3.2. Comparison of results. How authors compared the results? Did they use some software with other algorithms or did they program everything on their own?
  5. The paper would be more valuable if the authors would give some practical examples.
  6. Can this method adopt for other PID controlled systems?
  7. Comparison with previously published work should be more given in more detail [13]. Have you compared to some other work and another PID controlled system?
  8. The paper should be written according to the guide for authors (e.g. equation marks (2.) ?…).

The reviewer thinks that the paper can be resent for review if the author reasonably answers these questions.

Author Response

See the attached file please.

Reviewer 2 Report

I have read with interest the article "Optimal control of chilled water system based on improved sparrow search algorithm". 

This is a well-documented and well-written manuscript that can be published in the journal of Buildings, still, there should be paid close attention to the text - see line 225 - table 2. The paper is very comprehensible and very abundant in data, also the conclusion reflects the research work.

Author Response

See the attached file please.

Reviewer 3 Report

I do not recommend this paper for publication in the current version. Here are the main comments and suggestion:

1- The simulation results are not sufficient/comprehensive and the authors should extensively examine the proposed approach for different scenarios. 

2- The proposed approach should be compared with the sate-of-the-art algorithms and the current simulation results are not convincing. 

3- A section "Related methods" can make the paper easier to follow.

4- The main contributions of the authors should be explicitly mentioned in the introduction.

5- There is not any organization part in the last paragraph of the introduction.

6- The detailed software/hardware used to implement the algorithms should be reported in the paper.

Author Response

See the attached file please.

Round 2

Reviewer 1 Report

The author has made all the necessary changes or answered the question well.
Before publishing, paper should be checked by english native reader.

Reviewer 3 Report

I recommend the paper in the current form for publication!

This manuscript is a resubmission of an earlier submission. The following is a list of the peer review reports and author responses from that submission.